# Brain-Derived Neurotrophic Factor (BDNF) Enhances Osteogenesis and May Improve Bone Microarchitecture in an Ovariectomized Rat Model

**DOI:** 10.3390/cells13060518

**Published:** 2024-03-15

**Authors:** Eugene J. Park, Van-Long Truong, Woo-Sik Jeong, Woo-Kie Min

**Affiliations:** 1Department of Orthopedic Surgery, Kyungpook National University Hospital, College of Medicine, Kyungpook National University, Daegu 41566, Republic of Korea; osjpark@knu.ac.kr; 2Food and Bio-Industry Research Institute, School of Food Science & Biotechnology, College of Agriculture and Life Sciences, Kyungpook National University, Daegu 41566, Republic of Korea; truong88@knu.ac.kr

**Keywords:** bone formation, brain-derived neurotrophic factor, osteoblastic differentiation

## Abstract

Background: Brain-derived neurotrophic factor (BDNF) has gained attention as a therapeutic agent due to its potential biological activities, including osteogenesis. However, the molecular mechanisms involved in the osteogenic activity of BDNF have not been fully understood. This study aimed to investigate the action of BDNF on the osteoblast differentiation in bone marrow stromal cells, and its influence on signaling pathways. In addition, to evaluate the clinical efficacy, an in vivo animal study was performed. Methods: Preosteoblast cells (MC3T3-E1), bone marrow-derived stromal cells (ST2), and a direct 2D co-culture system were treated with BDNF. The effect of BDNF on cell proliferation was determined using the CCK-8 assay. Osteoblast differentiation was assessed based on alkaline phosphatase (ALP) activity and staining and the protein expression of multiple osteoblast markers. Calcium accumulation was examined by Alizarin red S staining. For the animal study, we used ovariectomized Sprague-Dawley rats and divided them into BDNF and normal saline injection groups. MicroCT, hematoxylin and eosin (H&E), and tartrate-resistant acid phosphatase (TRAP) stain were performed for analysis. Results: BDNF significantly increased ALP activity, calcium deposition, and the expression of osteoblast differentiation-related proteins, such as ALP, osteopontin, etc., in both ST-2 and the MC3T3-E1 and ST-2 co-culture systems. Moreover, the effect of BDNF on osteogenic differentiation was diminished by blocking tropomyosin receptor kinase B, as well as inhibiting c-Jun N-terminal kinase and p38 MAPK signals. Although the animal study results including bone density and histology showed increased osteoblastic and decreased osteoclastic activity, only a portion of parameters reached statistical significance. Conclusions: Our study results showed that BDNF affects osteoblast differentiation through TrkB receptor, and JNK and p38 MAPK signal pathways. Although not statistically significant, the trend of such effects was observed in the animal experiment.

## 1. Introduction

Bone is a continuously renewed tissue that undergoes a dynamic remodeling throughout the life course. Bone homeostasis is maintained by the balance between osteoblast-mediated bone formation and osteoclast-mediated bone resorption [1]. Osteoclasts are large multinucleated cells that are differentiated from hematopoietic myeloid precursors and exhibit bone-resorbing activity via degradation of the bone matrix [2]. Osteoblasts stemming from the bone marrow mesenchymal stem cells and located in the bone remodeling units are responsible for the formation of bone extracellular matrix by producing bone matrix components, such as collagen type 1, alkaline phosphatase (ALP), osteopontin (OPG), osteocalcin, osteomodulin, and osteonectin [3,4]. An imbalance between bone formation and resorption due to reduced proliferation and osteoblastic differentiation of bone marrow-derived stromal cells results in reduced bone regeneration. Osteoporosis is one of the most common age-related bone diseases characterized by a reduction of bone mass, deterioration of bone tissues, and disruption of bone microstructure, leading to an increased risk of fractures [5]. Inhibition of osteoclastic activity and/or stimulation of osteoblast differentiation is one of the most common and effective strategies for developing novel osteoporosis medications [2].

Recent studies have reported that various factors originating from the nervous system can directly and/or indirectly regulate bone metabolism, homeostasis, and remodeling [6,7,8]. Neurotransmitters released from the nervous system, such as substance P, calcitonin gene-related peptide, and neuropeptide Y, play an important role in regulating the regeneration, motility, proliferation, and differentiation of bone marrow cells [9,10,11,12]. In particular, increased release of bone injury-induced neurotransmitters vitally contributes to bone healing [13,14]. Pre-clinical and clinical studies have observed that brain-derived neurotrophic factor (BDNF) level is enhanced following traumatic brain injury, possibly involving fracture healing [15,16,17,18,19].

BDNF, belonging to the family of neurotrophins, is often released from the central and peripheral neuronal tissues and essentially functions in the nervous systems, such as neuronal development, survival and differentiation, neurogenesis, synaptic plasticity, and cognitive function [20,21]. Recent studies have shown that BDNF has positive effects on bone formation by enhancing osteoblast differentiation of human mesenchymal stem cells [22]. In addition, BDNF also stimulates MC3T3-E1 cell differentiation and promotes new bone formation and maturation [18]. Furthermore, suppression of BDNF expression by using long-coding RNA BDNF-antisense inhibited osteogenesis differentiation of bone marrow mesenchymal stem cells, possibly through inverse regulation of BDNF and osteogenic signaling pathway [23].

Although several reports evaluated the effect of BDNF on osteoblast differentiation, the molecular mechanism of its action, aiming at essential components of the signal transduction machinery that plays a central role in the differentiation process, has not been fully investigated. Therefore, this study was conducted to investigate the osteogenic potential and underlying mechanism of BDNF on bone marrow-derived stromal cells. Furthermore, since the protective effect of BDNF against osteoporosis in estrogen-deficient rat models remains unknown, the effect of BDNF on bone remodeling was also examined in an ovariectomized-induced osteoporosis rat model.

## 2. Materials and Methods

### 2.1. Materials

BDNF was purchased from R&D Systems (Minneapolis, MN, USA). Ascorbic acid, β-glycerophosphate, and neutral formalin were purchased from Sigma-Aldrich (St. Louis, MO, USA). Anti-ALP, OPN, BMP2/4, Osterix/Sp7, β-actin, and HRP-conjugated anti-mouse secondary antibodies were purchased from Santa Cruz Biotechnology (Santa Cruz, CA, USA). Anti-Runx2, p-ERK, ERK, p-JNK, JNK, p-p38, p38, TrkB, and HRP-conjugated anti-rabbit secondary antibodies were acquired from Cell Signaling Technology (Beverly, MA, USA). Anti-p-TrkB antibody was obtained from Abcam (Cambridge, UK).

### 2.2. Cell Culture

MC3T3-E1 subclone 4, a mouse preosteoblast cell line, was purchased from the American Type Culture Collection (Manassas, VA, USA) and cultured in α-Minimum Essential Medium without ascorbic acid (α-MEM, Gibco, Gaithersburg, MD, USA) containing 10% fetal bovine serum (FBS, Gibco, Gaithersburg, MD, USA) and 1% antibiotic (100 µg/mL penicillin and 100 U/mL streptomycin, Gibco, Gaithersburg, MD, USA) in a humidified CO_2_ incubator at 37 °C. ST-2, a mouse bone marrow stromal cell, was obtained from RIKEN Cell Bank (Tsukuba Scientific City, Ibaraki, Japan) and maintained in RPMI-1640 medium (Gibco, Gaithersburg, MD, USA) supplemented with 10% FBS and 1% antibiotic (100 µg/mL penicillin and 100 U/mL streptomycin) in a humidified atmosphere of 95% air with 5% CO_2_ at 37 °C. During culture, fresh media were replaced every three days.

For osteogenic differentiation, the 70% confluent cells were incubated in the osteogenic induction medium (OIM), including α-MEM medium plus 10% FBS, 1% antibiotics, 10 mM β-glycerophosphate, and 50 µg/mL ascorbic acid. Bone modeling is regulated by the interaction of various cell types; among them, osteoblasts and bone marrow stromal cells can communicate with each other during bone formation. Thus, the effect of BDNF on osteogenic differentiation under the more closely physiological conditions of the MC3T3-E1 and ST-2 co-culture system was examined. For co-culture, a direct 2D co-culture system of MC3T3-E1 and ST-2 was maintained in α-MEM medium containing 10% FBS and 1% antibiotics in a humidified incubator under 5% CO_2_ balanced air at 37 °C. The media were changed every 2–3 days during the experiments.

### 2.3. Cell Proliferation Assay

To evaluate the effects of BDNF on cell proliferation, the cells were seeded in 96-well plates (5000 cells/well) and treated with various concentrations of BDNF (10–200 ng/mL) for 1 and 3 days. The cell proliferation was then determined by the Cell Counting Kit-8 (CCK-8) assay kit (Sigma-Aldrich, St. Louis, MO, USA) according to the manufacturer’s instructions. The absorbance of the resulting solution was measured at 450 nm using a microplate reader (BioTek, Winooski, VT, USA).

### 2.4. Alkaline Phosphatase (ALP) Activity and Staining

The ALP activity was detected by a colorimetric assay using p-nitrophenyl phosphate as substrate. Briefly, the cells were seeded in 6-well plates (at a density of 1 × 10^4^ cells/cm^2^) and treated with different concentrations of BDNF for 3, 5, or 7 days. After treatment, the cells were lysed with assay buffer, and the total protein contents of the supernatants were measured by a PierceTM BCA protein assay kit (Thermo Fisher Scientific, Waltham, MA, USA). The ALP activity was determined by an ALP assay kit (Biovision, Waltham, MA, USA) according to the manufacturer’s guidelines and normalized to the protein content.

ALP staining was performed to confirm the results of ALP activity by using a 5-bromo-4-chloro-3-indolyl-phosphate (BCIP)/nitro-blue tetrazolium (NBT) solution (Sigma-Aldrich, St. Louis, MO, USA). After treatment, the medium was discarded, and the cells were washed twice with phosphate buffer saline (PBS, Sigma-Aldrich, St. Louis, MO, USA). The cells were fixed with 10% neutral formalin for 10 min at room temperature. The cells were then stained with BCIP/BNT solution for 30 min at 37 °C in the dark. Representative images were captured by a digital microscope camera (Paxcam, Iowa, IL, USA).

### 2.5. Mineralization Assay

Calcium accumulation was determined by an Alizarin Red S (Cell Biologics Inc., Chicago, IL, USA) staining. In brief, the cells were cultured in 12-well plates (at a density of 1 × 10^4^ cells/cm^2^) and then treated with various concentrations of BDNF for 14 or 21 days. After removing the medium, the cells were fixed with 10% neutral formalin for 10 min and rinsed with distilled water. The cells were then stained with 2% Alizarin Red S for 30 min at room temperature, followed by washing five times with distilled water. The representative images were captured using a digital microscope camera (Paxcam).

### 2.6. Western Blot

The cells were cultured in 6-well plates (at a density of 1 × 10^4^ cells/cm^2^) and treated with various concentrations of BDNF (10–200 ng/mL) for 3, 5, or 7 days. After washing twice with ice-cold PBS, the cells were lysed in RIPA buffer (Cell Signaling) containing fresh protease and phosphatase inhibitors (Sigma-Aldrich, St. Louis, MO, USA). Cell homogenates were centrifuged at 13,000 rpm for 10 min at 4 °C, and the protein concentration of resulting supernatants was quantified using a PierceTM BCA protein assay kit (Thermo Fisher Scientific). Equal amounts of proteins were separated in SDS-PAGE gels and then transferred to Immobilon^®^-P PVDF membranes (Merck Millipore, Burlington, MA, USA). After blocking with 5% skim milk in TBST (0.1% Tween 20 in Tris-buffered saline) overnight, the membranes were hybridized with indicated primary antibodies overnight at 4 °C and followed by incubation with proper horseradish-conjugated secondary anti-mouse or anti-rabbit antibody for 3 h at 4 °C. The protein bands were visualized with EzWestLumi plus reagent (Atto, Tokyo, Japan) using a LuminiGraph II system (Atto).

### 2.7. Animal Experiment

Twelve-week-old female Sprague-Dawley (SD) rats were purchased and housed in individual cages under specific pathogen-free conditions at 22 ± 2 °C and relative humidity of 55 ± 5% with a 12/12 h light–dark cycle. Throughout the experiment, rats were allowed free access to standard chow and water ad libitum. All animal experiments were approved by the Institutional Animal Care and Use Committee of Kyungpook National University (Protocol No. KNU 2022-0154, approved on 3 May 2022). A total of six ovariectomized (OVX) rats were prepared by removing both ovaries to produce an osteoporosis model. We divided the rats into two groups; the control group was injected with normal saline, and the BDNF group was injected with BDNF. BDNF was intraperitoneally injected twice a week starting from twenty-eight days after ovariectomy. The initial rat weight range was 210~250 g. The average weight was 200 g due to weight loss during the four weeks after ovariectomy. We administered BDNF 10 µg in the peritoneum weekly, in line with our preliminary previous experiment concentration (5 µg/100 g) for 12 weeks. After the 4 weeks of injection, due to weight gain (280~300 g), we have increased BDNF dosage to 15 µg for the next 8 weeks. At the end of the experiment, the rats were euthanized, and the bone structure was sectioned and stored for micro-CT and histological analysis.

#### 2.7.1. Micro-Computed Tomography

Specimens were scanned using a micro-computed tomography system (SkyScan 1173; Bruker Micro-CT, Kontich, Belgium). Scanning was made at 130 KV and a 1.0 mm aluminum filter was used. The image resolution was 19.97 µm. The images were reconstructed using NRecon software (Ver. 1.7.4.6, Bruker-microCT, Kartuizersweg 3B 2550, Kontich, Belgium). Acquired reconstructed images were sorted using Dataviewer (Ver. 1.5.6.2, Bruker-microCT, Kartuizersweg 3B 2550 Kontich, Belgium) and parameters were calculated using CtAn Software (Ver. 1.19.4.0, Bruker-microCT, Kartuizersweg 3B 2550 Kontich, Belgium). The midcoronal section of the proximal femur was chosen for investigation of the bone structure. The region of interest was the trabecular portion of the metaphysis between 0.5 mm and 1.5 mm distal to the physis. Bone mineral density was used for bone structure evaluation [24].

#### 2.7.2. Histomorphometry Analysis

Samples were fixed in 10% neutral buffered formalin. After dehydration, the Microtome (Leica RM2255 Fully Automated Rotary Microtome, Leica Biosystems Division of Leica Microsystems Inc., 1700 Leider Lane Buffalo Grove, IL, USA) was used for 3-micrometer slicing. De-paraffin using xylene, dehydration with 100%, 95%, and 70% alcohol, and washing was carried out. Afterwards, hematoxylin (dako hematoxylin) and eosin (BBC Biochemical), and tartrate-resistant acid phosphatase (TRAP) (Sigma-Aldrich, St. Louis, MO, USA) staining were performed to evaluate the osteoblast and osteoclast activity in both control and BDNF group. TRAP stain solution mix was placed in a staining dish and was pre-warmed to 37 °C in a water bath. Then, slides were de-paraffinized and rehydration was performed through graded ethanol to distilled water. Afterwards, slides were placed in a pre-warmed TRAP stain solution mix and incubated at 37 °C for 30 min or until control was developed. After rinsing in distilled water, we counterstained with 0.02% Fast Green for 30 s and a rinsing in distilled water was performed. Finally, it was dehydrated quickly through graded alcohols, 5 s each, clear in Xylene, and was mounted. Microscopic analysis was performed using Olympus BX51 (Olympus, Tokyo, Japan). A digital slide scanner (Pannoramic 250 Flash III, 3DHISTECH, Budapest, Hungary) was used to obtain images and analyzed using a viewing program (Caseviewer, 3DHISTECH, Budapest, Hungary).

### 2.8. Statistical Analysis

Data were presented as the mean ± standard deviation (SD) of at least three independent experiments and analyzed using GraphPad Prism 10 software (GraphPad Software Inc., La Jolla, CA, USA). Statistically significant differences were examined using the Mann–Whitney test, and one-way or two-way analysis of variance, followed by Tukey’s post hoc test. * *p* < 0.05, ** *p* < 0.01, and *** *p* < 0.001 values were considered statistically significant.

## 3. Results

### 3.1. Effect of BDNF on the Proliferation of ST-2 Cells

To investigate the proliferative effect of BDNF on bone marrow stromal cells, ST-2 cells were treated with various concentrations of BDNF (10–200 ng/mL) for up to 3 days, and cell proliferation was examined using a CCK-8 assay kit. Compared to the control group, BDNF did not significantly affect cell proliferation after 1 and 3 days of treatment (Figure 1).

### 3.2. Effect of BDNF on the Osteoblast Differentiation of ST-2 Cells

ALP plays an important role in the mineralization of newly formed bone and increases during cell proliferation and differentiation. Thus, ALP assay is widely used to screen bioactive compounds on bone formation. Firstly, the effect of BDNF on osteoblast differentiation of bone marrow stromal cells was analyzed by measuring ALP activity, which is a marker for the early and middle stages of osteoblastic differentiation. Results showed that BDNF (10–200 ng/mL) did not produce a significant effect on ALP activity after 3 and 5 days of treatment. However, when compared to the OIM group, BDNF significantly increased the ALP level at day 7. Of the tested concentrations, the highest effect on ALP activity was observed following the 50 and 100 ng/mL of BDNF treatment, whereas ALP activity was reduced at 200 ng/mL BDNF (Figure 2A). These results were confirmed by ALP staining, which showed high ALP-stained levels in BDNF-treated groups (Figure 2B). In addition, BDNF treatment also increased the expression of osteogenesis-related markers, such as osteopontin (OPN), osterix/Sp7, bone morphogenetic protein 2 (BMP2), and runt-related protein 2 (Runx2), but reduced the level of RANKL (Figure 2C).

Calcium deposition in the extracellular matrix is a phenotypic marker of the final stages of osteoblast differentiation. Therefore, to evaluate whether BDNF induces calcium accumulation, ST-2 cells were incubated with various concentrations of BDNF (10–200 ng/mL) for 21 days and subjected to Alizarin red S staining assay. Results showed that calcium accumulation was considerably increased by BDNF treatment at concentrations ranging from 10 to 200 ng/mL. Treatment with 100 ng/mL BDNF showed the highest calcium accumulation was observed in 100 ng/mL BDNF, while calcium deposition was slightly reduced by the treatment of 200 ng/mL BDNF (Figure 2D).

### 3.3. Effect of BDNF on the Mitogen-Activated Protein Kinase (MAPK) Signaling Pathways in ST-2 Cells

To investigate the effect of BDNF on MAPK signaling pathways in bone marrow stromal cells, the activation of MAPKs, including extracellular signal-regulated kinase (ERK), c-Jun N-terminal kinase (JNK), and p38 MAPK, was examined. As shown in Figure 3A, BDNF (100 ng/mL) treatment significantly activated JNK and p38 MAPK, and slightly induced ERK from 15 min to 1 h, as evidenced by increased phosphorylation. The phosphorylated levels of ERK, JNK, and p38 MAPK did not change after 3–6 h of BDNF treatment.

To further investigate whether MAPKs are involved in BDNF-induced osteoblast differentiation in bone marrow stromal cells, the effects of specific MAPK inhibitors, including PD98059 (ERK inhibitor), SP600125 (JNK inhibitor), and SB203580 (p38 MAPK inhibitor), in the presence and absence of BDNF (100 ng/mL) on osteoblast differentiation in ST-2 cells, were examined. As shown in Figure 3B,C, the ERK inhibitor (PD98059) highly increased ALP activity, whereas the p38 MAPK inhibitor (SB203580) remarkably reduced ALP activity both in OIM- and in BDNF-treated ST-2 cells. Additionally, the JNK inhibitor (SP600125) was unlikely to affect ALP activity in OIM-treated ST-2 cells but partially reversed the effect of BDNF. Furthermore, BDNF-induced increases in ALP, OPN, BMP2, BMP4, Osterix/Sp7, and Runx2 protein expression were obviously decreased by the JNK and p38 MAPK inhibitors. The effect of the p38 MAPK inhibitor was greater than that of the JNK inhibitor. In contrast, the ERK inhibitor did not affect BDNF-induced gene expression (Figure 3D). Moreover, BDNF-induced calcium deposition was strongly inhibited by the p38 MAPK inhibitor, whereas the JNK inhibitor did not affect the matrix mineralization (Figure 3E). These findings suggest that BDNF stimulates osteoblast differentiation in bone marrow stromal cells mainly through p38 MAPK and at least in part via JNK signaling pathways.

### 3.4. TrkB Suppression Reduces BDNF-Induced Osteoblast Differentiation

To examine whether BDNF could activate the TrkB receptor in bone marrow stromal cells, ST-2 cells were treated with BDNF (100 ng/mL), and the phosphorylation of TrkB was determined at various indicated times. As shown in Figure 4A, BDNF rapidly activated the TrkB receptor in bone marrow stromal cells, as indicated by increased protein phosphorylation after 5 min of treatment. The activation of the TrkB receptor continuously remained until 1 h and gradually reduced after 3 h of BDNF treatment.

To understand the role of the TrkB receptor in BDNF-induced osteogenesis, the effects of the TrkB receptor inhibitor (K252a) in the presence and absence of BDNF on osteoblast differentiation of ST-2 cells were examined. Results showed that the TrkB receptor inhibitor (K252a) significantly reversed the effect of BDNF on ALP activity (Figure 4B,C). In addition, K252a considerably reduced the expression of osteogenic markers, such as ALP, OPN, BMP2, BMP4, Osterix/Sp7, and Runx2, in the BDNF-treated ST-2 cells (Figure 4D). Furthermore, BDNF-promoted calcium accumulation was remarkably inhibited in the presence of a TrkB receptor inhibitor (Figure 4E). These results suggest that BDNF promotes osteogenic differentiation of bone marrow stromal cells through TrkB receptor activation.

### 3.5. Effect of BDNF on the Osteogenic Differentiation in the Co-Culture System of Osteoblasts and Bone Marrow Stromal Cells

BDNF (50 and 100 ng/mL) significantly increased the ALP activity in the co-culture system after 5 and 7 days of treatment (Figure 5A,B). In addition, the protein expressions of osteogenic markers, including ALP, OPN, osterix/Sp7, and Runx2, were upregulated by BDNF treatment (Figure 5C). Furthermore, BDNF enhanced matrix mineralization in the co-culture system, as indicated by increased levels of calcium deposition (Figure 5D).

To further evaluate whether MAPKs play a role in BDNF-induced osteogenic differentiation in co-culture systems, the effects of PD98059 (ERK inhibitor), SP600125 (JNK inhibitor), and SB203580 (p38 MAPK inhibitor) in the presence and absence of BDNF (100 ng/mL) on osteogenic differentiation of MC3T3-E1 and ST2 co-culture system were examined. ALP staining showed that ERK inhibitors strongly enhanced ALP levels both in OIM- and BDNF-treated groups after 5 days of treatment. In contrast, BDNF-induced ALP levels in the co-culture system were attenuated by the JNK and p38 MAPK inhibitors (Figure 5E). Similarly, the p38 MAPK inhibitor strongly inhibited BDNF-induced calcium deposition, whereas the ERK inhibitor enhanced this event in the co-culture system. BDNF-induced calcium accumulation was partially prevented following treatment with a JNK inhibitor (Figure 5F). These findings support the above observations that BDNF promotes osteoblast differentiation mainly through p38 MAPK and partially via the JNK signaling pathway.

### 3.6. BDNF May Affect Bone Density in OVX-Induced Rats

#### 3.6.1. Micro-Computed Tomography

Micro-CT analysis showed that the bone mineral density of the BDNF group was higher compared to the control group but statistically not significant (Figure 6).

#### 3.6.2. Histomorphometry Analysis

In the HE stain, the osteoblast surface/bone surface ratio was higher in BDNF compared to the control group; however, it did not show statistical significance. In the TRAP stain, BDNF showed a significantly lower osteoclast surface/bone surface ratio compared to the control group. (Figure 7).

## 4. Discussion

Due to the rapidly aging population worldwide, osteoporosis has become a critical public health issue, affecting millions of people, especially the elderly and postmenopausal women [5]. Its pathological process is complex and current mainstream treatment options have limitations. Thus, an urgent need is to identify safe anabolic agents that can effectively promote bone formation on a long-term basis to compensate for increased bone resorption and to improve bone mass in patients with osteoporosis. Recent studies have shown that numerous neurotransmitters, such as substance P, nerve growth factor, BDNF, neurotrophin 3 (NT-3), and neurotrophin 4 (NT-4), are implicated in bone formation and fracture healing [24,25,26]. Among them, BDNF, belonging to the family of neurotrophins, has attracted great attention due to its extensive bioactivity [15,16,18,19].

Recently, BDNF has been reported to stimulate osteoblast differentiation of osteoblast-linage cell MC3T3-E1 and human bone mesenchymal stem cells and enhance the expression of osteogenesis-related markers, such as ALP and osteocalcin [18,27]. BDNF also upregulates the expression of bone/cementum-related proteins, including ALP, OPN, and BMP-2 in cementoblasts [28]. However, the effects of BDNF on osteoblast differentiation in bone marrow stromal cells and osteoporotic conditions, as well as its detailed mechanisms, have not been fully investigated. In this study, we focused on the investigating effects of BDNF on osteoblast differentiation in stromal cells and co-culture systems and associated bone formation signaling via MAPKs and TrkB receptors. We found that BDNF promoted osteoblast differentiation via the activation of the TrkB receptor along with the JNK and p38 MAPK signaling pathways. In vivo, this study failed to show significant improvement in bone density compared to the control group. We assume that the limited sample size led to such a result, although the BDNF group showed higher bone density, a higher osteoblast number, and a lower osteoclast number. A future study with a larger sample size focusing on bone microarchitecture and bone turnover markers is necessary to conclude our preliminary findings in OVX rats.

Bone marrow stromal cells, non-hematopoietic multipotent cells capable of differentiating into mesodermal cell types such as osteoblast and adipocytes, play an important role in the normal development of bone [29]. Osteoporosis occurrence may be associated with the imbalance of bone marrow stromal cells, which differentiate toward adipocytes instead of osteoblastic fate [30]. Osteoblasts stemming from bone marrow stromal cells experience a precise program of gene expression to regulate physiological processes, including osteoblastic commitment, proliferation, and terminal differentiation [31]. In this study, BDNF did not affect the proliferation of ST-2 cells. This finding is in line with previous studies that BDNF had no proliferative effect on human bone mesenchymal stem cells [27], as well as MC3T3-E1 osteoblast [18]. In contrast, BDNF has been found to enhance the proliferation of MLO-Y4 osteocytes [32]. Taken together, these observations reveal that the effects of BDNF on proliferation vary with the tested cell line.

Although BDNF did not change ST-2 cell proliferation, it obviously induced the osteoblast differentiation and maturation processes. BDNF treatment significantly increased ALP activity, a marker of the early phase of osteoblast differentiation [31]. Moreover, BDNF considerably increased the matrix mineralization, as indicated by increased levels of calcium deposition. These findings suggest that BDNF has positive effects on osteoblast differentiation through enhanced cellular differentiation and mineralization in bone marrow-derived stromal cells, without affecting cell proliferation.

During the osteoblast differentiation and mineralization of bone marrow-derived stromal cells, these cells drive a program of gene expression governed by various transcription factors [33]. Runx2 is regarded as the master transcription factor that promotes the differentiation of mesenchymal progenitors into osteoblasts by regulating the expression of osteogenesis-related genes at the early stage of osteoblastic differentiation, such as ALP, collagen alpha 1 type I, and osteocalcin [31,34]. Osterix/Sp7 plays a crucial role at the later step in the osteoblast differentiation process, that is, the differentiation of pre-osteoblasts into mature osteoblasts and osteocytes [1]. In this study, BDNF significantly increased Runx2 and Osterix/Sp7 expression in ST-2 cells and co-culture system, implying that BDNF is able to promote osteogenesis in bone marrow stromal cells throughout all phases of osteoblast differentiation and bone formation. Bone morphogenetic proteins, multi-functional factors that belong to the transforming growth factor beta superfamily, also play an important role in the osteoblastic differentiation of mesenchymal stem cells and bone formation by targeting Runx2 expression [35]. BMP2 expression was enhanced by BDNF treatment in ST-2 cells and a co-culture system. In addition, BDNF simultaneously increased OPG expression and reduced RANKL levels, suggesting that BDNF promotes osteogenic bone formation via elevating the OPG/RANKL ratio. OPG, produced by osteoblasts as well as osteocytes, plays a suppressive role in RANKL-induced osteoclastogenesis [36]. Thus, the OPG/RANKL ratio may be considered an ultimate determinant of bone integrity and reflect the balance between bone formation and resorption [37].

To gain further insights into the mechanisms of BDNF in promoting osteogenic differentiation, we first examined the MAPK signaling molecules. The ERK, JNK, and p38 MAPK signaling pathways are believed to be associated with osteoblast differentiation [38,39,40]. It has been reported that ERK signaling is required for osteoblast differentiation and osteogenic marker gene induction in human bone mesenchymal stem cells [27]. In addition, ERK is involved in BDNF-induced vascular endothelial growth factor in rat osteoblasts [41]. In contrast, BDNF was found to induce the Akt signal but did not affect the ERK signal during mouse MC3T3-E1 osteoblast differentiation [18]. This study observed that BDNF activated JNK and p38 MAPK signaling pathways, but not ERK, during osteoblast differentiation in ST-2 cells. The discrepancy in these results may be due to the different cell lines used in the experiment, and the effect of BDNF varies with tested cell lines. Furthermore, JNK and p38 MAPK inhibitors diminished BDNF-induced osteoblast differentiation both in ST-2 cells and the co-culture system. Our results demonstrated that JNK and p38 MAPK inhibitors, but not ERK inhibitors, abolished the effects of BDNF on ALP activity and expression of osteogenesis-related markers, such as ALP, OPN, BMP2/4, Runx2, and Osterix/Sp7. The p38 MAPK inhibitor appeared to be more effective in suppressing BDNF-induced osteoblast differentiation and matrix mineralization than JNK inhibitors. Taken together, these findings indicate that BDNF-induced osteogenesis is mediated mainly by p38 MAPK and partially via JNK signaling pathways.

Recent studies have indicated that the effect of BDNF on osteoblast differentiation may be related to its receptor TrkB [19,27,29,41,42]. Both BDNF and its TrkB receptor are required at various stages of the bone formation process and are associated with fracture healing [19,29]. In this study, we found that BDNF treatment early activated TrkB receptors in bone marrow-derived stromal cells. Furthermore, osteoblast differentiation was significantly reduced in the presence of a TrkB inhibitor. The suppression of the TrkB receptor significantly reduced BDNF-induced ALP activity, as well as the expression of osteogenesis-related markers, such as ALP, OPN, BMP2/4, Runx2, and Osterix/Sp7. In addition, BDNF-induced matrix mineralization was prevented by TrkB inhibitor. These findings suggest that the TrkB receptor is required for BDNF-mediated osteogenesis in bone marrow-derived stromal cells.

## 5. Conclusions

This study demonstrates that BDNF promotes osteoblast differentiation and mineralization in bone marrow-derived stromal cells. BDNF-induced osteogenesis is mediated mostly by the TrkB receptor via JNK and p38 MAPK signaling pathways. Moreover, the in vivo study supported that BDNF may provide a protective effect against ovariectomy-induced bone loss. It is assumed that the positive influence of BDNF on osteoblast differentiation in bone marrow stromal cells may, at least in part, contribute to such a protective effect. These results provide additional evidence to extend the existing information on the therapeutic effect of BDNF for the prevention and/or treatment of osteoporosis.

## Figures and Tables

**Figure 1 cells-13-00518-f001:**
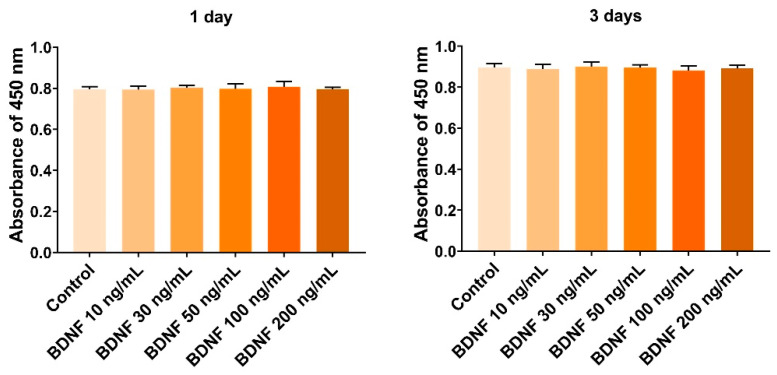
Effect of BDNF on proliferation of bone marrow-derived stromal ST-2 cells. The cells were treated with various concentrations of BDNF (10–200 ng/mL) for 1 and 3 days. The cell proliferation was measured using the CCK-8 assay kit.

**Figure 2 cells-13-00518-f002:**
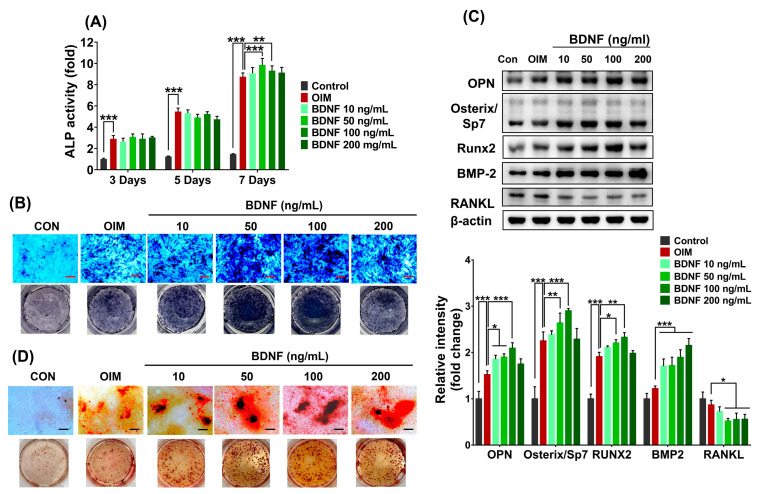
Effect of BDNF on osteoblast differentiation in ST-2 cell. The cells were treated with BDNF (10–200 ng/mL) in osteogenic induction medium (OIM) for 3, 5, 7, or 21 days. (**A**) ALP activity on days 3, 5, and 7. (**B**) ALP staining on day 7. (**C**) Protein expressions of OPN, Osterix/Sp7, Runx2, BMP-2, and RANKL on day 7. (**D**) Calcium deposition on day 21. Scale bar = 400 µm. Data are presented as the mean ± SD of three to five independent experiments. The values of * *p* < 0.05, ** *p* < 0.01, and *** *p* < 0.001 are considered as statistically significant differences.

**Figure 3 cells-13-00518-f003:**
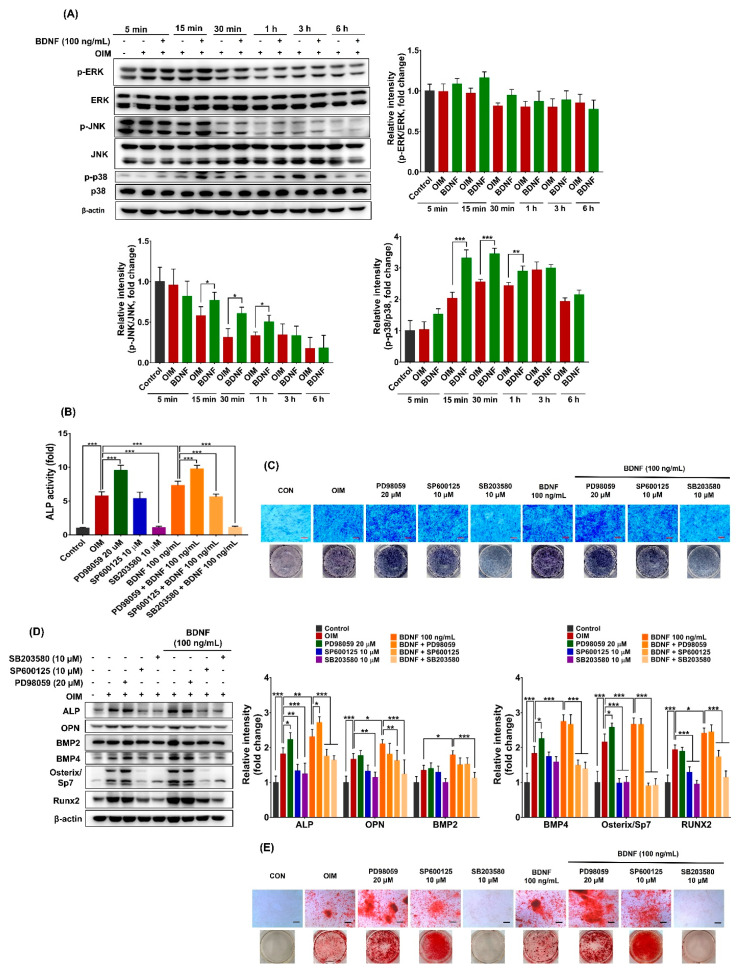
Involvement of MAPK signaling pathways in BDNF-induced osteoblast differentiation in ST-2 cells. (**A**) The cells were treated with BDNF (100 ng/mL) in OIM for 5 min, 15 min, 30 min, 1 h, 3 h, and 6 h, and protein expressions of p-ERK, ERK, p-JNK, JNK, p-p38 MAPK, and MAPK were determined by Western blot. (**B**–**D**) The cells were treated with PD98059 (ERK inhibitor, 20 µM), SP600125 (JNK inhibitor, 10 µM), and SB203580 (p38 MAPK inhibitor, 10 µM) in the presence and absence of BDNF (100 ng/mL) for 7 or 21 days. (**B**) ALP activity on day 7. (**C**) ALP staining on day 7. (**D**) Protein expressions of osteogenic markers, including ALP, OPN, BMP2, BMP4, Osterix/Sp7, and Runx2 on day 7. (**E**) Calcium deposition on day 21. Scale bar = 400 µm. Data are presented as the mean ± SD of three to five independent experiments. The values of * *p* < 0.05, ** *p* < 0.01, and *** *p* < 0.001 are considered as statistically significant differences.

**Figure 4 cells-13-00518-f004:**
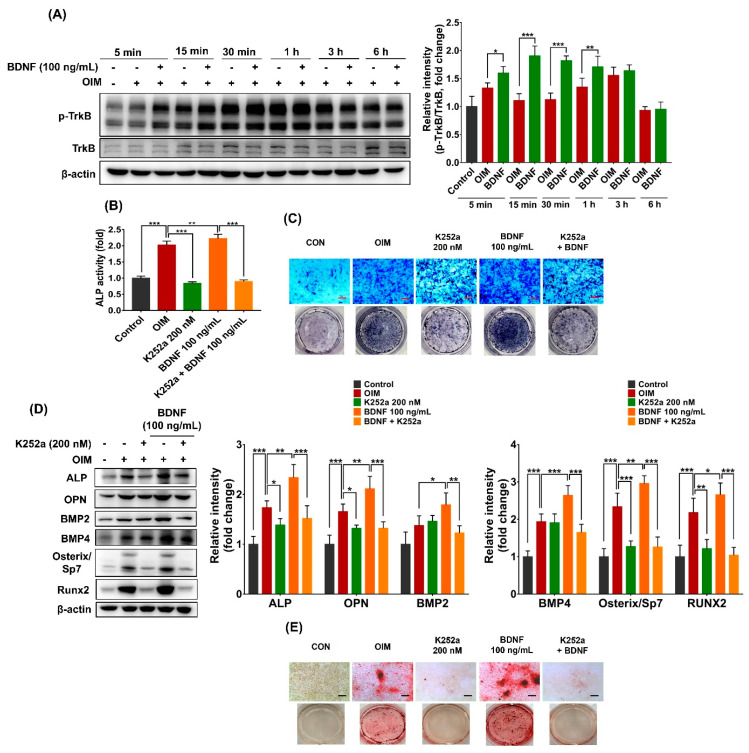
Involvement of TrkB receptor in BDNF-induced osteoblast differentiation in ST-2 cells. (**A**) The cells were treated with BDNF (100 ng/mL) in OIM for 5 min, 15 min, 30 min, 1 h, 3 h, and 6 h, and protein expressions of p-TrkB and TrkB were determined by Western blot. (**B**–**D**) The cells were treated with K252a (TrkB inhibitor, 200 nM) in the presence and absence of BDNF (100 ng/mL) for 3, 5, 7, or 21 days. (**B**) ALP activity on days 3, 5, and 7. (**C**) ALP staining on day 7. (**D**) Protein expressions of osteogenic markers, including ALP, OPN, BMP2, BMP4, Osterix/Sp7, and Runx2 on day 7. (**E**) Calcium deposition on day 21. Scale bar = 400 µm. Data are presented as the mean ± SD of three to five independent experiments. The values of * *p* < 0.05, ** *p* < 0.01, and *** *p* < 0.001 are considered as statistically significant differences.

**Figure 5 cells-13-00518-f005:**
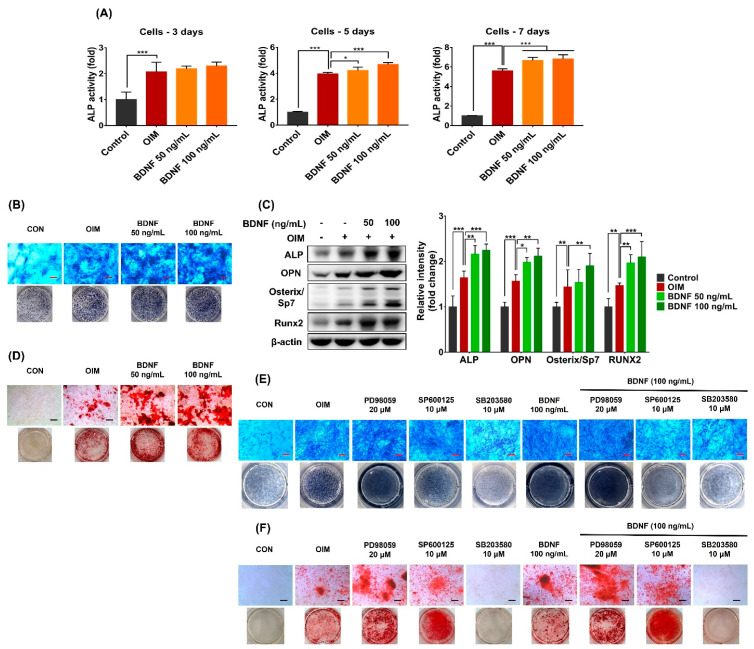
Effect of BDNF on osteoblast differentiation in co-culture system of ST-2 and MC3T3-E1 cells. (**A**–**D**) The co-culture systems were treated with BDNF (50 or 100 ng/mL) for 3, 5, 7, or 21 days. (**A**) ALP activity on days 3, 5, and 7. (**B**) ALP staining on day 5. (**C**) Protein expressions of ALP, OPN, BMP2, and Runx2 on day 5. (**D**) Calcium deposition on day 21. (**E**,**F**) The co-culture systems were treated with PD98059 (ERK inhibitor, 20 µM), SP600125 (JNK inhibitor, 10 µM), and SB203580 (p38 MAPK inhibitor, 10 µM) in the presence and absence of BDNF (100 ng/mL) for 5 or 21 days. (**E**) ALP staining on day 5. (**F**) Calcium deposition on day 21. Scale bar = 400 µm. Data are presented as the mean ± SD of three to five independent experiments. The values of * *p* < 0.05, ** *p* < 0.01, and *** *p* < 0.001 are considered as statistically significant differences.

**Figure 6 cells-13-00518-f006:**
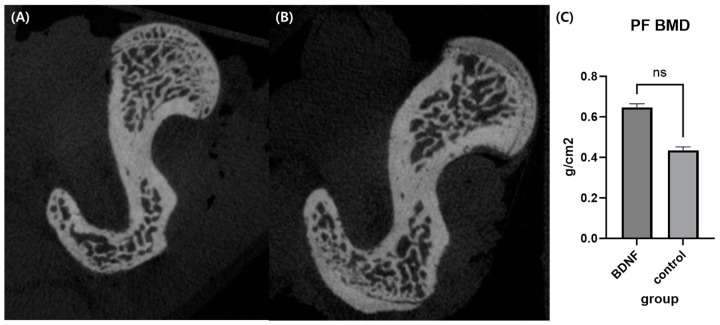
Micro-CT analysis. Representative axial cuts of proximal femur of BDNF (**A**) and control group (**B**). (**C**) Comparison of bone mineral density between BDNF and control group. ns, non-specific; PF, proximal femur; BMD, bone mineral density.

**Figure 7 cells-13-00518-f007:**
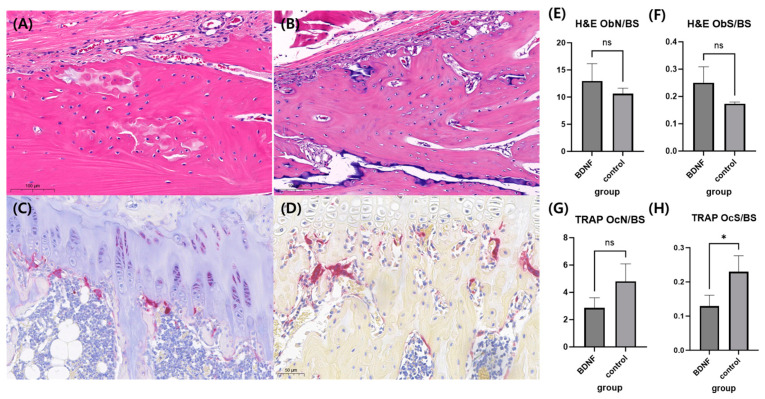
Histologic analysis. (**A**) HE stain in BDNF group. (**B**) HE stain in control group. (**C**) TRAP stain in BDNF group. (**D**) TRAP stain in control group. (**E**,**F**) Comparison of osteoblast number/bone surface ratio and osteoblast surface/bone surface ratio between BDNF and control group. (**G**,**H**) Comparison of osteoclast number/bone surface ratio and osteoclast surface/bone surface ratio between BDNF and control group. The value of * *p* < 0.05 is considered a statistically significant difference. ns; non-specific; ObN, osteoblast number; BS, bone surface; ObS, osteoblast surface; OcN, osteoclast number; OcS, osteoclast surface.

## Data Availability

Data are available on request due to privacy/ethical restrictions.

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
