# Peer review of "Brain-Derived Neurotrophic Factor (BDNF) Enhances Osteogenesis and May Improve Bone Microarchitecture in an Ovariectomized Rat Model"

_cells, 2024, doi:10.3390/cells13060518_

Round 1

Reviewer 1 Report

Comments and Suggestions for Authors

The authors present a well prepared manuscript detailing extensive in vitro experiments that clearly demonstrate the stimulatory effect of BDNF on osteogenesis, providing promising insights into future therapeutic approaches to degenerative bone disease. The in vitro work employs well-established cell culture methods to clearly show an effect of BDNF on osteoblast differentiation and function, and to interrogate specific signaling mechanisms. The animal studies appear to lack the statistical significance to demonstrate that this osteogenic effect translates in vivo and is protective of induced osteoporosis, and would require further results to be featured and interpreted in the manuscript as they currently are.

Specific Comments/Questions

1) The abstract appears to not mention the rat model at all, only the cell line experiments?

2) The introduction does a good job outlining the gap in mechanistic knowledge, but the final paragraph could be better organized/worded to setup up the rationale/novelty of experimental approaches used in this study. For example following the first sentence, highlight the in vitro experiments employed to investigate specific signaling mechanisms, then point to the importance of demonstrating that the osteoblast stimulation has a protective effect against osteoporosis, using the rat model.

3) An explanation of the rationale behind the experimental design using both cell types separately and in combination in direct co-culture should be included earlier in the manuscript, perhaps briefly in the introduction and/or methods sections, differentiating between progenitor and osteoblast cell lines for the reader.

4) Details regarding biological, technical, and experimental replicates should be further clarified. From the methods section and figure legends, the data is presented as 'mean +/- standard deviation of at least three separate experiments':

a. Please clarify if this referring to individually run experiments from different passages/plating of the cell lines.

b. If so, please clarify how the data shown in figures are pooled across these experiments - are you taking a mean of your replicate wells for each group as a value for that experiment, and then taking the mean of those values across multiple experiments?

c. The number of replicate wells per experiment should be specified as well, perhaps in the individual experiment methods subsections or in the statistical analysis subsection if they are all the same.

d. From the figure legends, it is unclear why different groups were pooled from different numbers (3-5) of separate experiments, were all groups not included in all experimental repeats, or some timepoints not repeated as many times?

5) I realize it is how the CCK-8 assay reports it, but to me the y-axis label in Figure 1 of 'cell viability %' is somewhat confusing in the context of measuring proliferation as it does not indicate total cell number - your experimental group could have proliferated 10-fold more, but if half die the viability could be 50% compared to a 100% viable control with less cells. Unless you are actually comparing the absorbance to readings from the starting timepoint and plating density to calculate the percent of viable cells, perhaps just briefly describe it as cell number, as directly related to absorbance in the methods/figure legend.

6) In general some of the large panel figures (2-5) are difficult to read, fitting in a lot of plots and images at a small scale and low resolution. Some elements could be resized to better take up the space, or just make a larger overall figure, or perhaps some figures could be split up.

7) What is the sample size for the animal experiments? This should be stated in the methods as well as represented as individual points in the plots for Figures 6 and 7.

8) It's slightly misleading in Figure 6 to set the y-axis range right to start right at the control value, thereby making the difference between the two groups seem obvious even though they are not statistically different.

9) In general the animal study data appears to be underpowered perhaps, with no mention of the sample sizes or experimental repeats, and no significant differences other than osteoclast number. Given this, the interpretation of these results in the discuss (Lines 367-369) are overstated. The TRAP data does support the decrease in RANKL observed in vitro so still worth including, but perhaps as a supplement rather than featured so directly in the manuscript.

Comments on the Quality of English Language

Only a few minor errors found which the editorial team will surely address.

Reviewer 2 Report

Comments and Suggestions for Authors

In this manuscript from Park and colleagues, the authors investigate the effects of BDNF on osteogenic cells. To achieve this, in most of their experiments they focused on treating ST2 bone marrow stromal cells under osteogenic culture conditions, showing that BDNF enhanced osteogenic differentiation. Additional data suggest that BDNF’s effects are mediated through TrkB and MAP Kinase pathways. Finally, they tested BDNF’s effects in a rat post-ovariectomy (OVX) bone loss model, reporting that BDNF attenuated bone loss. The cell culture aspects of the study were well designed and well reported, but significant deficiencies around the OXV studies need to be addressed.

The overall idea that BDNF promotes osteoblast differentiation is has been established in prior studies so it’s not entirely novel. I believe that the ST2 stromal cell model has not been previously examined for sensitivity to BDNF, so this is somewhat new. Greater novelty is with their mechanistic investigations, which suggest effects are mediated via TrkB receptors and MAP kinases. These studies are introduced, data presented, and the results interpreted well enough. 

            Unfortunately, there are numerous concerns around the rat post-OVX bone loss study. This experiment is poorly incorporated into the manuscript. It isn’t mentioned at all in the Abstract and doesn’t really fit in the Introduction. The text of the Results that present the OVX model are very sparse and are not adequate. Finally, it’s only slightly mentioned in the Discussion section (lines 367-69), which states there was “improved bone turnover rate”; rates of bone formation or resorption were never actually measured, so this isn’t justified by their data. As the authors know, post-OVX bone loss is a complex multifactorial effect involving changes to both bone resorption and bone formation. There really isn’t any consideration of BDNF and bone loss or osteoclasts, which includes an absence of consideration of the literature looking at direct BDNF effects on osteoclasts. This includes recent papers from Xue, eLife 2021 and older papers from Sun, Int J Cancer 2012 and Ai, PLOS One 2012 that tested BDNF effects of osteoclast cultures. Moreover, Xue tested effects of a BDNF mimetic in the rat OVX model. The authors need to make more clear the experimental rationale, what’s the precise question being tested, and how their current work relates to the existing literature. 

Perhaps the biggest issue with this experiment is that to understand effects of BDNF on post-OVX bone loss, a sham-surgery control must be included, so that we could see how much bone is present at baseline, how much was lost due to the OVX, and then how much of the bone loss was relieved by BDNF. Since there was no sham control, it’s impossible to say how much bone loss was prevented, i.e. is it reduced by 50% or only reduced by 1-2%? Moreover, almost none of the differences reported in figures 6 or 7 reach statistical significance, so the import of their data are unclear. The Y-axis scales for these graphs are highly compressed, which may give an inaccurate impression how large or small are the differences between groups.

As lesser concerns, the Methods section for microCT and histomorphometry needs to give more detail around the ROI used for analyses. In Figure 6, it needs to be clearly indicated which micrograph represents the control or BDNF bone. Was BMD the only bone parameter analyzed from their microCT data? As a suggestion, serum biomarkers for bone formation and resorption would have further strengthened the interpretation of these studies.

Reviewer 3 Report

Comments and Suggestions for Authors

abstract:

1. In the material and method part, the author did not mention animal experiments.

Method:

1. The use of inhibitors should be described in detail in the method.

Results:

1. Figure 1 only provides the diagram of BNDF promoting cell proliferation on the 1st and 3rd day respectively, but does not provide the effect of BNDF promoting cell proliferation in the same concentration range, so the third diagram should be given.

2. Each picture in Figure 2 and figure 3 is too small. Please enlarge the picture size or combine the pictures again.

3. The part of animal experiment lacks the verification of molecular pathway results.

Comments on the Quality of English Language

It is suggested that the content of the article be modified and polished.

Reviewer 4 Report

Comments and Suggestions for Authors

The reviewer recommends that the authors address the following:

1. Section 2.7.1 and section 3.6.1.

1.1 Authors fail to provide information about how many specimens were scanned and analyzed by microtomography, please provide details.

1.2 Authors do not provide information about dataset calibration to determine bone mineral density. 

1.3 Why do authors only performed analysis of bone mineral density? It is reductive and do not provide details on the bone microstructural features. A full analysis of cortical and trabecular bone should be performed, which includes, at least, the following parameters: cortical thickness, cortical porosity, trabecular thickness and trabecular separation. 

1.4 Authors do not indicate which volume of interest was used for microtomographic analysis.

2. Section 2.7.2

2.1 What do authors mean by "exclusive viewing program"?

2.2 Please provide details on method for TRAP staining.

2.3 Please provide details about how histomorphometrical analysis, through image, was performed.

3. Results regrading in vivo assay are not entirely acknowledged in the discussion section. Please further discuss the relevance of your findings within the aim of the study and regarding translational conclusions.

Comments on the Quality of English Language

No major issues in this regard are detected.

Reviewer 5 Report

Comments and Suggestions for Authors

 The title, "Brain-Derived Neurotrophic Factor (BDNF) Enhances Osteogenesis and Positively Affects Bone Structure in Ovariectomized Rat Model," appears to deviate from the primary focus of the study, which centrally involves the investigation of signaling pathways within in vitro cell cultures. A more accurate title could better highlight the study's core emphasis on exploring signaling pathways in the context of osteogenesis using cell lines.

The plagiarism detection program has identified a 37% similarity (already excluding the preprint), surpassing the acceptable threshold of 25%. It is acknowledged that some repetition may occur in methods and figure legends, but efforts should be made to minimize such occurrences. Special attention is required for the abstract and conclusion sections, where similarity exceeds 50% and necessitates substantial revisions.

Abstract

The abstract's title suggests a focus on the impact of BDNF on the bone structure of osteoporosis-induced rats, but this aspect is not explicitly addressed in the abstract itself.

The primary emphasis of the work should be on investigating the signaling pathway, especially with the utilization of inhibitors. However, a significant portion of the abstract discusses basic aspects of BDNF's osteogenic effects. It would be more appropriate to reduce the details of these basic effects in the background and methods sections, allowing for a more detailed exploration of the critical signaling pathway in the abstract.

Introduction

The introduction is lacking the critical idea that led to the extensive work performed in the study; it merely touches the surface of the topic. I recommend condensing the first two paragraphs and delving into the role of BDNF in osteogenesis, highlighting potential pathways. This approach will provide a more comprehensive background, explaining why specific signaling pathways and inhibitors were selected for the study and the rationale behind using a combination of two cell lines. Additionally, clarifying the relevance of the ovariectomized rat model and its connection to the pathways in the context of estrogen-deprivation-induced osteoporosis would enhance the introduction and tide in the work that has been done nicely.

Method

1. Correct the CO2 to CO(subscription)

2. Provide additional information on the cell seeding density employed in each experiment and the duration of their culture in the well plates before commencement of the experiments. Moreover, clarity on the frequency of media changes during the experiments is essential. These details are crucial as cell density and secreted cytokines from the cell lines may significantly impact proliferation and osteogenesis outcomes.

3. Indicate how many rats were used for each group of the experiments.

Results

1. In Figure 4E, the image appears distorted; kindly maintain the original image ratio.

2. For Figure 6, isn't it essential to include a control representing normal bone?

3.To enhance visual interpretation, it is recommended to include scale bars in Figures 6 and 7. Additionally, provide specific annotations for elements such as A and B in (A-B) or E, F, G, H. Ensure that all abbreviations, including ObS, OcN, OcS, and BS, are thoroughly described for clarity. What are the error bar represents and *, ns meaning.

4. The title "3.6 BDNF Prevents Bone Loss in OVX-Induced Rats" may be misleading, as the results indicate no differences in bone mineral density (BMD), and there is no discernible change in the number of osteoblasts in histology.

5. Consider generate a concluding graphic summarizing the effects of BDNF from the study. Given the numerous names of inhibitors and signaling molecules in the results, a visual aid could enhance clarity and facilitate a more straightforward understanding of the key findings, thereby improving the overall manuscript.

6. Upon review, the data and statistical analysis of graph 6C indicate a potential statistical difference, contrary to being non-significant (ns) as initially suggested by the error bars. Verify the accuracy of the statistical results for this graph or the data that used to generate error bar of the graph.

Discussions

Something wrong with the in-text citation, please correct them all.

Discussion is informative and well written. However, the ovariectomized part was left off again.  The relation of the identified pathway and the role of BDNF in recovering bone loss in estrogen-deficiency should be discussed. What is the localization of BDNF, how intraperitoneal injection could help in reducing osteoclast activity or does it really “positively” effect bone structure as stated in the title?

Conclusion

Rewrite to minimize similarity to the previous works as suggested.

Comments on the Quality of English Language

Recheck for typos in texts and clarity in the figure legend.

Reviewer 6 Report

Comments and Suggestions for Authors

Manuscript of E.J. Park et al. “Brain-Derived Neurotrophic Factor (BDNF) Enhances Osteogenesis and Positively Affects Bone Structure in Ovariectomized Rat Model” describes the experimental study, which is aimed to investigate the action of BDNF on the osteoclast differentiation of bone marrow stromal cells. In addition, participation of MAP kinase signal transduction pathways is explored. The authors followed up effects of BDNF on proliferation and osteogenic differentiation of ST-2 bone marrow stromal cells (alone or in co-culture with MC3T3-E1 preosteoblast cell line), which was characterized by evaluating the alkaline phosphatase activity, osteogenic marker expression, and calcium deposition. Additionally, the role of MAP kinases ERK, JNK and p38 is evaluated by using selective inhibitors. In the animal model of ovariectomized rats, the influence of BDNP it was evaluated on bone mineralization, as well as osteoblast and osteoclast content.

As overall, authors demonstrated that BDNF, dependently on its receptor TrkB stimulates osteogenic differentiation of ST-2 bone marrow stromal cells with participation of JNK and p38 kinases. The osteogenic effects of BDNF were confirmed in the co-culture experiments as well as in animal model of ovariectomized rats.

Reviewer has no major has no major comments on this paper.

Minor comments and questions to authors.

1.       What is the specific role of ST-2 and MC3T3-E1 cells in the co-culture system, in addition to "more closely physiological conditions"? Whether some intercellular contact and/or paracrine mechanisms may be involved?

2.       What is the general conclusion from the data in Figures 6 and 7? The answer to this and previous questions should preferably be posted in the Discussion.

3.       p-TrkB/TrkB, pERK/ERK, pJNK/JNK, pp38/p38 are represented by two forms on western blots. Were they analyzed separately or together?

4.       It is necessary to enclose reference numbers in square brackets (Lines.355, 349, 356, 359, 361, 372, 374, 376, 378, …., 431)

5.       Please include full names and abbreviations of all items, like “osteoclast number/bone surface (OcN/BS)” and so on, in legends to figures 6 and 7

Comments on the Quality of English Language

Line 352: ...have showed... Please, check the text of the manuscript 

Round 2

Reviewer 3 Report

Comments and Suggestions for Authors

The author has answered the reviewer's questions perfectly, and I have no more questions.

  Comments on the Quality of English Language

The author has answered the reviewer's questions perfectly, and I have no more questions.

Author Response

Thank you for taking your time to review.

Reviewer 4 Report

Comments and Suggestions for Authors

The reviewer considers that following needs to be further addressed/detailed:

1. Regarding microtomographic analysis, the selected VOI encompasses only the trabecular portion. However the coverletter states that cortical bone was assessed and no differences were found. Can the authors clarify?

2. The details added to section 2.7.2 refer to sample preparation to histological analysis, it does not addressed how TRAP staining was performed. 

3.  Regarding histomorphometrical analysis, it is not clear if the presented quantative analysis was performed by manual counting or using automated/semi-automated methods. In case of the latter, briefly explain the method.

Comments on the Quality of English Language

No major issues were detected.

Round 3

Reviewer 3 Report

Comments and Suggestions for Authors

The author has made a good revision on the basis of the original text, and I have no unnecessary questions.

Reviewer 4 Report

Comments and Suggestions for Authors

The reviewer has no futher questions.